# Time-variant Duo-image Inpainting via Interactive Distribution Transition Estimation

## Abstract

In this work, we focus on a novel and practical task, *i.e.*, Time-vAriant duo-iMage inPainting (TAMP). The aim of TAMP is to inpaint two damaged images by leveraging their complementary information, where the two images are captured at the same scene with a significant time gap between them, *i.e.*, time-variant duo-image. Different from existing reference-guided image inpainting, TAMP considered the potential pixel damage and content mismatch of reference images when they are collected from the Internet for real-world applications. In particular, our study finds that even state-of-the-art (SOTA) reference-guided image inpainting methods fail to address this task due to inappropriate image complementation. To address the issue, we propose a novel Interactive Transition Distribution Estimation (ITDE) module that interactively complements the duo-image with semantic consistency and provides refined inputs for the consequent image inpainting process. The designed ITDE is inpainting pipeline independent making it a plug-and-play image complement module. Thus, we further propose the Interactive Transition Distribution-driven Diffusion (ITDiff) model, which integrated ITDE with a SOTA diffusion model, as our final solution for TAMP. Moreover, considering the lack of benchmarks for TAMP task, we newly assembled a dataset, *i.e.*, TAMP-Street, based on existing image and mask datasets. We conduct experiments on our TAMP-Street and conventional DPED50k datasets which show our methods consistently outperform SOTA reference-guided image inpainting methods for solving TAMP.

## 1 Introduction

Image inpainting (Lugmayr et al., 2022; Zhang et al., 2023; Wang et al., 2022c) refers to the task of restoring a damaged image, *i.e.*, the target image, to its original content. Currently, conventional image inpainting methods (Guo et al., 2021; Li et al., 2022c) still struggle to recover the target images when the pixel missing is extensive or the semantic structures are complex. To tackle such "ill-posed" problem, reference-guided image inpainting (Oh et al., 2019), *i.e.*, RefInpaint, was proposed where another image of the same scene, *i.e.*, the reference image, is introduced and served as inpainting guidance. With extra prior knowledge, reference-guided image inpainting methods (Zhou et al., 2021; Li et al., 2022b; Liao et al., 2023; Cao et al., 2024) have achieved promising results and more practical applications. However, existing reference-guided image inpainting studies are limited by assuming the reference images are complete and highly similar to the damaged target images, potentially leading to failure when the reference and the target images exhibit serious discrepancies.

As illustrated in Fig. 1, the reference images $\mathbf{I}_1$ of existing studies, *e.g.*, Zhou et al. (2021), Liao et al. (2023) and Cao et al. (2024), are usually set as intact and have no significant difference to the target images $\mathbf{I}_2$. Generally, these images are collected in the same period of time, *e.g.*, two different frames from a single video clip like Zhou et al. (2021). As a result, existing reference-guided image inpainting researches mainly focus on handling the misalignment, *i.e.*, the style and geometric

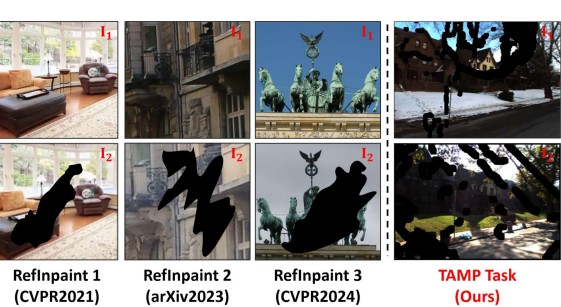

**Figure 1:** Samples from existing RefInpaint studies and ours.

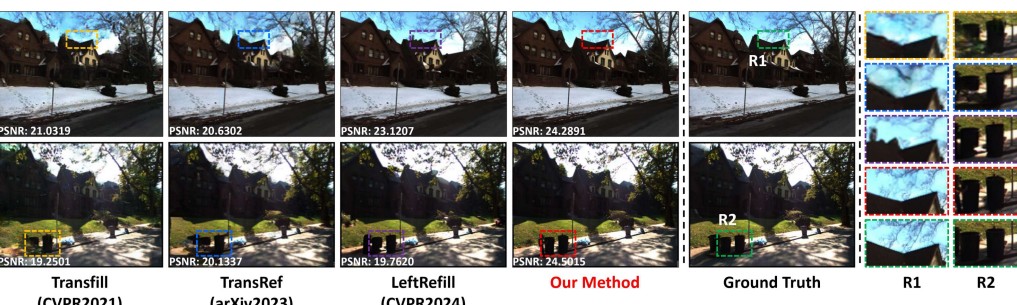

**Figure 2:** The inpainting results of existing RefInpaint methods and our method for the TAMP case in Fig. 1.

difference, between the two images. However, as claimed in these existing studies (Zhou et al., 2021; Liao et al., 2023; Cao et al., 2024), the reference images should be collected from the Internet for real application concerns, which means a large time gap inevitably exists between the reference and target images. For example, there may be a seasonal change between the two images as illustrated in Fig. 1, where the content mismatch now becomes the critical challenge rather than the misalignment. Moreover, the reference image may also suffer from damage due to the existence of a large time gap. Therefore, in this work, we propose the Time-vAriant duo-iMage inPaint (TAMP) task by considering the reference images with damage and having significant content mismatch to the target images.

Intuitively, one can stick to existing reference-guided image inpainting methods for realizing TAMP. Nevertheless, as shown in Fig. 2, it is clear that even the SOTA method cannot achieve plausible results. In general, we note that appropriate image complementation is of critical importance for realizing TAMP, yet existing methods require a dedicated complementation strategy. Therefore, we explicitly designed a module, *i.e.*, Interactive Transition Distribution Estimation (ITDE), to interactively complement the two images. In specific, the proposed ITDE considers the complementation as a distribution transition process between the two images. By interactively exchanging features and predicatively filtering semantics (Li et al., 2022c) during the transition process, ITDE is capable of only complementing semantically consistent contents. To facilitate existing methods to utilize the complemented results, we adopt two output heads, *i.e.*, complement and confidence heads, where the complement head outputs the interactively complemented images and the confidence head gives out the confidence of the complementation results. The two heads provide optimized inputs for subsequent inpainting pipelines, thereby boosting the performance of existing models for TAMP task. In summary, this study contributes to the image inpainting research with following contributions.

- We present the Time-vAriant duo-iMage inPainting (TAMP) task which further enhances the practical applications of existing reference-guided image inpainting research by considering the reference images suffering from damage and having significant differences to the target images.

- To address the TAMP task, we design a novel module, Interactive Transition Distribution Estimation (ITDE), to complement the images without introducing significant artifacts.

- Benefit to the plug-and-play nature of ITDE, we further propose the Interactive Transition distribution-driven Diffusion (ITDiff) as our final TAMP solution.

- We build the TAMP-Street dataset to evaluate SOTA and propose methods for solving TAMP. The comprehensive experiments prove the superiority of the proposed ITDE and ITDiff.

## 2 RELATED WORK

**Traditional image inpainting** mainly focus on the designing of different inpainting pipelines, *i.e.*, single-shot, two-stage and progressive method. The single-shot methods essentially learn a mapping from a corrupted image to the completed one, *e.g.*, mask-aware design (Zhu et al., 2021; Wang et al., 2021b;a), attention mechanism (Zhang et al., 2022; He et al., 2022; Zheng et al., 2022a) and many others (Zeng et al., 2022; Lu et al., 2022; Feng et al., 2022; Wang et al., 2022b). As for two-stages methods, coarse-to-fine (Kim et al., 2022; Roy et al., 2021) and structure-then-texture (Yamashita et al., 2022; Wu et al., 2021) are two main adopted methodologies. Some further works (Zeng et al., 2020; Li et al., 2022a) extend the two-stages method and propose to inpaint image progressively, where the image holes are iteratively completed from the boundary to the center. Parallel to the

pipeline designing, generative model is also popular for image inpainting research. Researchers initially utilize VAE (Kingma & Welling, 2013) and GAN (Goodfellow et al., 2014) as the backbone to generate missing contents for the corrupted image (Zheng et al., 2021; 2022b). Later, some researches also adopt flow-based methods (Wang et al., 2022a) and masked language models (Wan et al., 2021) for image inpainting tasks. Most recently, diffusion models (Ho et al., 2020) with dedicated sampling strategy designing (Lugmayr et al., 2022; Zhang et al., 2023; Wang et al., 2022c) have become the main utilized generative model and have achieved superior inpainting performance.

**Reference-guided image inpainting** was recently proposed to tackle the failing of traditional methods where the holes are large or the expected contents have complicated semantic layout. With extra reference images introduced, those methods (Zhou et al., 2021; Liu et al., 2022; Li et al., 2022b; Liao et al., 2023; Cao et al., 2024) manage to achieve more visually convincing results. Specifically, Zhou et al. (2021) propose a multihomography fusion pipeline combined with deep warping, color harmonization, and single image inpainting to address the issue of parallax between the target and reference images. Later Liu et al. (2022) proposes to separately infer the texture and structure features of the input image considering their pattern discrepancy of texture and structure during inpainting. More recently, Liao et al. (2023) introduced a transformer-based encoder-decoder model to better harmonize the style differences, and construct a publicly accessible benchmark dataset containing 50K pairs of input and reference images. The most up-to-date method (Cao et al., 2024) adopts the powerful Visual Language Model (VLM) for further boosting the performance and has set up new SOTA for the reference-guided image inpainting task.

## 3 Time-variant Duo-image Inpainting

To discuss TAMP with concreteness, we first formulate the problem and introduce the challenges in Sec. 3.1. Next in Sec. 3.2, we intuitively resort to SOTA reference-guided image inpainting method, *i.e.*, LeftRefill (Cao et al., 2024), for a case study. Based on the results, we determine the cause of the poor performance which motivates us to propose our TAMP solution, *i.e.*, ITDE and ITDiff, in Sec. 4.

### 3.1 Problem Statement

We consider two images captured from the same scene but with a significant time gap between them, *i.e.*, time-variant duo-image $(\mathbf{I}_1, \mathbf{I}_2)$. Different from traditional inpainting setup, the duo-image of TAMP are both damaged with unknown pixels missing. Our aim is to recover the original content of the duo-image by taking each other as reference. Formally, we can formulate the problem as

$$(\hat{\mathbf{I}}_1, \hat{\mathbf{I}}_2) = \phi(\mathbf{I}_1, \mathbf{I}_2), \text{ s.t. } \mathbf{I}_1 = \text{Shoot}(\mathbf{S}, \mathbf{M}_1, t_1), \ \mathbf{I}_2 = \text{Shoot}(\mathbf{S}, \mathbf{M}_2, t_2), \ t_2 - t_1 \gg 0, \quad (1)$$

where $\mathbf{I}_i = \text{Shoot}(\mathbf{S}, \mathbf{M}_i, t_i)$ means capturing an image $\mathbf{I}_i$ of scene $\mathbf{S}$ at time stamp $t_i$ that damaged by pixel missing $\mathbf{M}_i$. We use $t_2 - t_1 \gg 0$ to indicate the significant time gap of the duo-image. $(\hat{\mathbf{I}}_1, \hat{\mathbf{I}}_2)$ is the reconstructed duo-image and $\phi(\cdot)$ is the desired function to achieve the goal.

Compared to conventional reference-guided image inpainting, the TAMP defined above raises several non-trivial challenges due to the large time gap. As shown by the rectangles in Fig. 3 (a), the main objective of existing reference-guided image inpainting is to solve the style and geometric misalignment between the two images. However, TAMP further faces the object mismatch and appearance change obstacles, as shown in Fig. 3 (b) (c), which not only requires image alignment but also proper content complementation. ❶ **Object Mismatch.** There are significant changes in objects between the duo-image of TAMP due

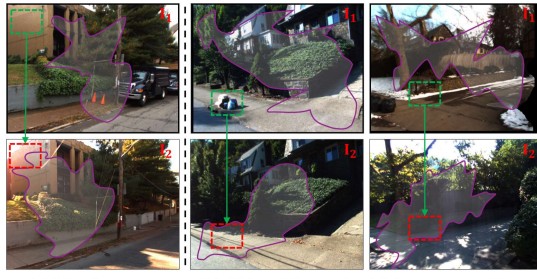

(a) Misalignment    (b) Object Mismatch    (c) Appearance Change

**Figure 3:** Conventional and our TAMP's challenges. The regions enclosed by purple lines refer to the pixel damages.

to the large time gap. For instance, as depicted in Fig. 3 (b), the object enclosed by the green rectangle in $\mathbf{I}_1$ vanishes in $\mathbf{I}_2$, indicated by the red rectangle. Consequently, what is initially an unlost object in $\mathbf{I}_1$ transforms into interference rather than serving as complementary cues when we attempt to reconstruct $\mathbf{I}_2$ based on $\mathbf{I}_1$. ❷ **Appearance Change.** Furthermore, the appearances of the same object

in the duo-image can also be distinct due to the weather changes and light variations. As it can be seen from Fig. 3 (c), the same road corner scene have totally different appearances because of the environmental variation. Even if there is no obvious object distinction, it is still hard to utilize the complementary pixels in $\mathbf{I}_1$ to recover $\mathbf{I}_2$. Such a fact further undermines the complementary cues and poses another challenge.

## 3.2 OBSERVATION AND MOTIVATION

Despite the challenges highlighted above, one can still directly apply existing reference-guided image inpainting methods to realize TAMP. However, we show in this section that even SOTA reference-guided inpainting method, *i.e.*, LeftRefill(Cao et al., 2024), cannot achieve convincing results. The details of model setup and more visualization results can be found in Appendix B.2.

❶ **Basic Inpainting.** We first directly apply LeftRefill to inpaint the target duo-image under the conventional reference-guided image inpainting setup. As the "Basic Results" visualized in Fig. 4, it is clear that LeftRefill fails to refill the duo-image with original content. Noticeably, as the red rectangles in the "Basic Results" indicated, the lion statues have a clear reference in the counterpart images but LeftReill fails to recover them. Empirically, such poor performance is due to the fail of explicitly utilizing the counterpart image for reference. Without considering the complementary information between the duo-image, such a pipeline may render the inpainting process incapable of preserving the image content authenticity. ❷ **Complemented Inpainting.** Therefore, we further test LeftRefill with naively complemented duo-image, which can be formulated as

$$\tilde{\mathbf{I}}_1 = \mathbf{I}_1 + \mathbf{I}_2 * \bar{\mathbf{M}}_1, \ \ \tilde{\mathbf{I}}_2 = \mathbf{I}_2 + \mathbf{I}_1 * \bar{\mathbf{M}}_2 \qquad (2)$$

where $\bar{\mathbf{M}}_i$ means the inverse of the binary mask. Fortunately, as the blue rectangles in "Comp. Masked GT" shown, the lion statue can be restored through naive image complementation and the "Comp. Results" successfully preserve its structure, *i.e.*, the green rectangle. Nevertheless, it can be observed that the results still exhibit significant artifacts for the masked regions and borders. This is due to the style and content discrepancy introduced by naive image complementation which greatly disturbed the inpainting process. Take the yellow rectangle as an example, the woody building frame should not exist in the final output which is definitely a hindrance for desirable inpainting.

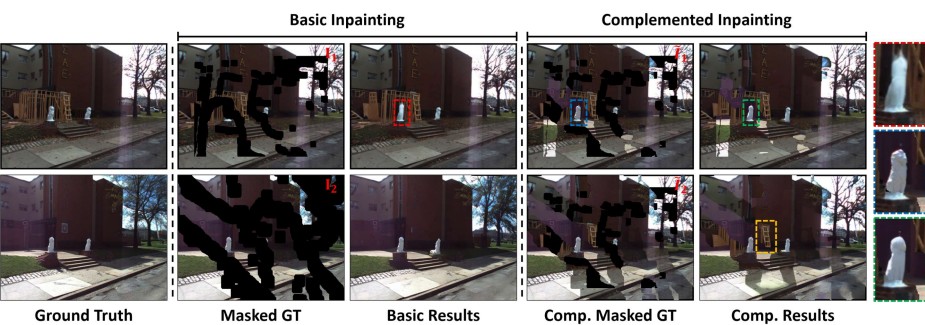

**Figure 4:** Inpainting results of LeftRefill for exemplar duo-image under basic and naive complementing setup.

In general, we can conclude from the above analysis that image complement is of critical importance for successful TAMP. However, naively copying and pasting the contents between the duo-image will introduce substantial artifacts which significantly impede the inpainting process. Thus in the next, we absorb our attention to the realization and utilization of appropriate image complementation strategy.

## 4 METHODOLOGY

Observing the discrepancy in naively complemented images, *i.e.*, $\tilde{\mathbf{I}}_1$ and $\tilde{\mathbf{I}}_2$ in Fig. 4, we note that *the semantic contradiction of complemented contents is the primary cause of the poor performance*. Such semantic contradictions are essentially introduced by the two new challenges we discussed in Sec. 3.1, which makes it the key issue for solving TAMP. Generally, what we aim to complement is the image regions with matching semantics, while those semantically contradicted regions should be suppressed to avoid interference. To achieve such desired image complementation, we consider the duo-image complementation as a distribution transition process rather than naive copy-and-paste.

**Figure 5:** The proposed Interactive Transition Distribution Estimation module. The data flow of duo-image are highlighted with orange and blue arrow lines. Mask $\mathbf{M}_i$ and confidence $\mathbf{C}_i$ are inversed for better visualization.

As Fig. 5 illustrated, we explicitly design a module, *i.e.*, Interactive Transition Distribution Estimation (ITDE), to learn such transitions. By assessing the semantic consistency during the distribution transition, ITDE ensures only the semantically matched contents are complemented between the duo-image. It is worth noting that ITDE itself is independent of the existing inpainting pipeline which makes it a general plug-and-play technique for solving TAMP tasks. Consequently, after the introduction of ITDE in Sec. 4.1, we propose the Interactive Transition distribution-driven Diffusion (ITDiff) as our final solution for solving TAMP, *i.e.*, Sec. 4.2.

## 4.1 INTERACTIVE TRANSITION DISTRIBUTION ESTIMATION

**Backbone.** As Fig. 5 illustrated, ITDE utilizes a UNet-style backbone (Ronneberger et al., 2015) to interactively learn the transition process between the duo-image. With the corrupted duo-image and corresponding masks, *i.e.*, $\mathbf{I}_i$ and $\mathbf{M}_i$, fed as input, we first exchange information between them with latent feature fusion, *i.e.*, the feature concatenation during encoding in Fig. 5. By further encoding the fused feature, both similar and contradictory contents of the duo-image are merged together which can be regarded as feature-level naive complementation. Drawing inspiration from the remarkable semantic structure capturing capability of the semantic predictive filtering (SPF) technique (Li et al., 2022c), we apply it to filter out inconsistent information by assessing the semantic consistency of the merged contents, which can be formulated as

$$\hat{\mathbf{F}}_i = \sum_{\mathbf{q} \in \mathcal{N}_{\mathbf{P}}} \mathbf{K}_{\mathbf{p}}^j[\mathbf{q} - \mathbf{p}]\mathbf{F}_i[\mathbf{q}], \quad i, j \in \{1, 2\}, i \neq j, \tag{3}$$

where $\mathbf{p}, \mathbf{q}$ are the coordinates of feature elements and the set $\mathcal{N}_{\mathbf{P}}$ contains $N^2$ neighboring pixels of $\mathbf{p}$. $\mathbf{F}_i$ is the encoding outputs and $\mathbf{K}_{\mathbf{p}}^j$ is the kernel for filtering the $\mathbf{p}$-th element of $\mathbf{F}_i$ via the neighboring elements, *i.e.*, $\mathcal{N}_{\mathbf{P}}$. By interactively applying semantic predictive filtering, we expect only the semantically consistent contents are complemented. Finally, we can get the correctly complemented features $\hat{\mathbf{F}}_i^c$ by feeding the $\hat{\mathbf{F}}_i$ to the following 8 ResBlocks and decoding process.

**Complement & Confidence Heads.** In general, the above backbone only ensures appropriate image complementation. As it has been discussed in Sec. 3.2, existing methods simply ignore the useful information, *i.e.*, the complemented contents in our situation, during inpainting. Therefore, a proper utilization of the complemented results is still required. As shown in Fig. 5, we further propose the complement & confidence heads to provide refined inputs for the consequent inpainting process. Specifically, we adopt the same filtering operation to get the final complemented images and the corresponding pixel-wise confidence of the complemented results,

$$\tilde{\mathbf{I}}_i = \sum_{\mathbf{q} \in \mathcal{N}_{\mathbf{P}}} \mathbf{K}_{C,\mathbf{p}}^j[\mathbf{q} - \mathbf{p}]\hat{\mathbf{F}}_i^c[\mathbf{q}], \quad \mathbf{C}_i = \sum_{\mathbf{q} \in \mathcal{N}_{\mathbf{P}}} \mathbf{K}_{M,\mathbf{p}}^j[\mathbf{q} - \mathbf{p}]\hat{\mathbf{F}}_i^c[\mathbf{q}]. \tag{4}$$

Coupled with the complemented images $\tilde{\mathbf{I}}_i$, the confidence map $\mathbf{C}_i$ informs the subsequent image inpainting model which part of the complemented contents should be reserved and which needs further inpainting. In fact, as we will see in the experiments, *i.e.*, Sec. 6.3, such a combination of head design generally ensures further suppression of inevitable content inconsistency thus reducing its negative impact on the final inpainted results.

## 4.2 Interactive Transition Distribution-driven Diffusion

Given the damaged duo-image and corresponding masks $\mathbf{I}_i, \mathbf{M}_i$, they are first fed into ITDE and follow the data flow in Fig. 5 to get $\tilde{\mathbf{I}}_i, \mathbf{C}_i$. Then the $\tilde{\mathbf{I}}_i, \mathbf{M}_i$ play as the images and masks that further fed into existing inpainting methods to get final restored results $\hat{\mathbf{I}}_i$. To demonstrate the practical utilization of the designed ITDE, we build on DDNM (Wang et al., 2022c) and propose the Interactive Transition Distribution-driven Diffusion (ITDiff). Following diffusion notation conventions, we use $\mathbf{x}_0^i$ to denote the final inpainted result $\hat{\mathbf{I}}_i$ and setup DDNM sampling process as

$$\mathbf{x}_{t-1}^i = \frac{\sqrt{\bar{\alpha}_{t-1}}\beta_t}{1 - \bar{\alpha}_t}\hat{\mathbf{x}}_{0|t}^i + \frac{\sqrt{\alpha_t}(1 - \bar{\alpha}_{t-1})}{1 - \bar{\alpha}_t}\mathbf{x}_t^i + \sigma_t\epsilon, \quad \epsilon \sim \mathcal{N}(0, \mathbf{1}), \tag{5}$$

where $\hat{\mathbf{x}}_{0|t}^i = \mathbf{A}^\dagger(\tilde{\mathbf{I}}_i \odot \mathbf{C}_i) + (\mathbf{1} - \mathbf{A}^\dagger\mathbf{A})\mathbf{x}_{0|t}^i$ is the DDNM modification for diffusion sampling. $\mathbf{x}_{0|t}^i$ denotes the diffusion estimated $\mathbf{x}_0^i$ at time-step $t$ and $\mathbf{A}, \mathbf{A}^\dagger$ is the degradation operator and corresponding sudo-inverse. To further enhance the complementation during the diffusion process, we follow Gao et al. (2022) and interactively apply low-pass filter $\phi_D(\cdot)$ to DDNM sampling:

$$\hat{\mathbf{x}}_{t-1}^i \leftarrow \mathbf{x}_{t-1}^i - \omega \nabla_{\mathbf{x}_t^i} ||\phi_D(\tilde{\mathbf{I}}_j \odot \mathbf{C}_j) - \phi_D(\hat{\mathbf{x}}_{0|t}^i \odot \mathbf{C}_j)||_2, \tag{6}$$

which essentially requires the generation of the duo-image to align with each other at every time step.

## 5 TAMP-Street Dataset

As discussed in Sec. 1, the existing datasets only focus on the misalignment problem making them inadequate for practical application. Therefore, we assemble the TAMP-Street[1] dataset based on existing images and masks to evaluate models under TAMP setup. For a clear comparison, we summarize the main difference between TAMP-Street and the data utilized by existing researches in Tab. 1. The visualization of TAMP-Street can be found in Appendix A.

**Table 1:** Comparison of TAMP-Street and the datasets of existing reference-guided image inpainting studies.

| Obstacle Dataset | Misalignment | Object Change | Appearance Change | Reference Damage |
|---|---|---|---|---|
| TransFill | ✔ | ✘ | ✘ | ✘ |
| TransRef | ✔ | ✘ | ✔ | ✘ |
| LeftRefill | ✔ | ✔ | ✘ | ✘ |
| TAMP-Street | ✔ | ✔ | ✔ | ✔ |

**Time-variant Duo-image.** As the setup detailed in Sec. 3.1, the duo-images of the TAMP task are required to be taken from the same scene with a large time gap between them. To meet such specifications, we adopt the images from VL-CMU-CD (Alcantarilla et al., 2018) as our image basis for dataset building. VL-CMU-CD was originally proposed for object changing detection which documents one year of diverse urban street transformations. Specifically, there are a total of 1,362 pairs of street view images all of which have a size of $1024 \times 768$. Following the original split, 816 pairs of images are utilized for training, 256 pairs of images for validation, and 290 pairs for testing.

**Irregular Mask for TAMP.** To simulate the random pixel damage, we build our mask pairs based on the irregular hole mask introduced by Liu et al. (2018). Originally, the masks are split by mask ratio from 0%-60% with 20% as step. Benefiting from the advancements of the generative model, the restoration of small pixel missing has become an effortless task (Zhang et al., 2023), *i.e.*, Appendix C. Thus we only adopt mask ratio 20%-60% for TAMP evaluation and set the step ratio as 10%. In detail, we randomly pair the masks under each ratio and totally build 1600 pairs of masks for testing (400 under each mask ratio), 5600 and 800 pairs of masks for training and evaluation respectively. Note that we build our mask pairs by only utilizing the testing mask set of Liu et al. (2018).

---

[1]https://drive.google.com/drive/folders/1nEs48ZX8-4pGzjqlsLNgEKjxhXqoPCRb?usp=sharing

# 6 EXPERIMENTS

## 6.1 EXPERIMENTAL SETUP

**Baselines.** We compare the proposed ITDiff with three reference-guided image inpainting methods, *i.e.*, TransFill (Zhou et al., 2021), TransRef (Liao et al., 2023) and LeftRefill (Cao et al., 2024). In specific, TransFill (Zhou et al., 2021) represents traditional registration pipeline for image inpainting, TransRef (Liao et al., 2023) adopts Transformer architecture and belongs to the learning-based image inpainting method, and LeftRefill (Cao et al., 2024) utilizes powerful text-to-image foundation model, *i.e.*, Stable Diffusion 2, as the backbone for reference-guided image inpainting.

**Datasets.** For the purpose of verifying the ITDE's generalizability, we also conduct evaluation on DPED50k (Liao et al., 2023) which is a conventional reference-guided image inpainting dataset. DPED50k contains 52,000 image pairs in total, where 50,000 pairs are utilized for training and the rest 2,000 pairs for testing. For the purpose of validation, we further randomly split 1,000 out of 50,000 pairs of images as validation set. We keep the mask setup for DPED50k the same as TAMP-Street as described in Sec. 5 for TAMP evaluation.

**Model Training & Testing.** We pretrain ITDE on the training split of the TAMP-Street and DPED50k dataset respectively, where we follow Nazeri et al. (2019) and train ITDE with four loss functions, *i.e.*, $L_1$ loss, GAN loss, Style loss and perceptual loss. As for the diffusion model, the checkpoint from guided-diffusion (Dhariwal & Nichol, 2021) is utilized, *i.e.*, 256x256_diffusion_uncond.pt. During testing, we follow the pipeline described in Sec. 4.2 to test our proposed ITDiff. Note that all the results of selected baselines are obtained by following the "Basic Inpainting" pipeline described in Sec. 3.2. We refer readers to Appendix B.3 for comprehensive ITDE and baseline training details.

**Metrics.** Following the convention of image inpainting studies, we conduct evaluation with four metrics which are peak signal-to-noise ratio (PSNR), structural similarity index (SSIM), perceptual similarity (LPIPS) and $L_1$. PSNR, SSIM, and $L_1$ measure the quality of the recovered image, and LPIPS measures the perceptual consistency between the recovered image and ground truth. We report all the metrics for the reference and target images in separate for a clear comparison, *e.g.*, $L_1^{\mathbf{I}_1}/L_1^{\mathbf{I}_2}$.

## 6.2 COMPARATIVE RESULTS

**Table 2:** The quantitative results of the baselines and our proposed ITDiff for TAMP task on TAMP-Street dataset. The overall best results are highlighted in **bold** font.

| Mask
Method | 20%-30% | 30%-40% | 40%-50% | 50%-60% | 20%-30% | 30%-40% | 40%-50% | 50%-60% |
|---|---|---|---|---|---|---|---|---|
| | | PSNR ↑ | | | | SSIM ↑ | | |
| TransFill | 19.3377/20.1026 | 17.2356/17.6748 | 15.1677/15.8720 | 13.1019/13.7011 | 0.7800/0.7721 | 0.7233/0.7319 | 0.6807/0.6925 | 0.5567/0.5583 |
| TransRef | 18.9164/22.7751 | 17.2486/20.9556 | 15.9129/19.2969 | 14.6797/16.2012 | 0.8322/0.8617 | 0.7623/0.8067 | 0.6896/0.7438 | 0.6157/0.6298 |
| LeftRefill | 23.1727/23.1517 | 21.6623/21.9740 | 19.7019/19.5421 | 15.2239/15.9228 | 0.8949/**0.9054** | **0.8613**/0.8487 | 0.7950/**0.8123** | 0.6962/0.6971 |
| **ITDiff** | **26.2677/26.5959** | **24.5230/24.7559** | **22.7704/23.3243** | **20.4491/20.9545** | **0.8977**/0.8959 | 0.8533/**0.8513** | **0.8060**/0.8040 | **0.7278/0.7210** |
| | | $L_1$ ↓ | | | | LPIPS ↓ | | |
| TransFill | 0.1840/0.1814 | 0.1930/0.1951 | 0.3031/0.2894 | 0.3877/0.3503 | 0.1126/0.1184 | 0.1587/0.1639 | 0.2045/0.2121 | 0.2669/0.2767 |
| TransRef | 0.0819/0.0458 | 0.1159/0.0658 | 0.1515/0.0897 | 0.1976/0.1559 | 0.1631/0.1293 | 0.2279/0.1809 | 0.2883/0.2337 | 0.3482/0.3307 |
| LeftRefill | 0.0851/0.0883 | 0.1340/0.1239 | 0.1746/0.1821 | 0.3034/0.2744 | 0.0283/0.0290 | 0.0533/0.0520 | 0.0860/0.0783 | 0.1002/0.0925 |
| **ITDiff** | **0.0268/0.0264** | **0.0380/0.0379** | **0.0517/0.0495** | **0.0759/0.0739** | **0.0177/0.0182** | **0.0284/0.0269** | **0.0386/0.0407** | **0.0869/0.0736** |

**Table 3:** The quantitative results of the baselines and our proposed ITDiff for TAMP task on DPED50k dataset. The overall best results are highlighted in **bold** font.

| Mask
Method | 20%-30% | 30%-40% | 40%-50% | 50%-60% | 20%-30% | 30%-40% | 40%-50% | 50%-60% |
|---|---|---|---|---|---|---|---|---|
| | | PSNR ↑ | | | | SSIM ↑ | | |
| TransFill | 21.5232/22.1119 | 18.5660/18.3195 | 17.7743/18.2417 | 15.6322/15.3806 | 0.8517/0.8591 | 0.5314/0.4990 | 0.7270/0.7420 | 0.4726/0.4711 |
| TransRef | 21.1612/26.6525 | 19.4944/24.6318 | 18.3366/22.8359 | 17.1906/19.4730 | 0.8179/0.8900 | 0.7454/0.8419 | 0.6759/0.7852 | 0.6150/0.6862 |
| LeftRefill | 26.5375/30.9959 | 23.8886/27.9177 | 21.5189/24.8386 | 17.8494/20.5472 | 0.8991/0.9206 | 0.8435/0.8738 | 0.7747/0.8099 | 0.6678/0.7120 |
| **ITDiff** | **30.1669/32.3953** | **28.3700/30.7487** | **26.8271/29.2565** | **23.5205/25.5622** | **0.9275/0.9342** | **0.8988/0.9073** | **0.8644/0.8772** | **0.7841/0.8017** |
| | | $L_1$ ↓ | | | | LPIPS ↓ | | |
| TransFill | 0.0285/0.0270 | 0.0857/0.0876 | 0.0542/0.0495 | 0.0924/0.0916 | 0.1323/0.1269 | 0.2809/0.2931 | 0.2401/0.2306 | 0.3952/0.4163 |
| TransRef | 0.0695/0.0321 | 0.0978/0.0474 | 0.1263/0.0657 | 0.1614/0.1151 | 0.1955/0.1109 | 0.2642/0.1579 | 0.3243/0.2087 | 0.2944/0.3736 |
| LeftRefill | 0.0687/0.0557 | 0.1107/0.0939 | 0.1694/0.1543 | 0.2566/0.2462 | 0.1047/0.1030 | 0.1438/0.1422 | 0.1878/0.1837 | 0.2672/0.2616 |
| **ITDiff** | **0.0195/0.0155** | **0.0275/0.0220** | **0.0362/0.0289** | **0.0593/0.0496** | **0.0157/0.0096** | **0.0249/0.0159** | **0.0370/0.0254** | **0.0642/0.0477** |

**Compare with Existing Methods.** As the results tabulated in Tab. 2 and Tab. 3, we compare the proposed ITDiff with three baselines for solving the TAMP task on TAMP-Street and DPED50k. We have following observations. ❶ Compare to the most competitive baseline, *i.e.*, LeftRefill, our ITDiff generally achieves better results under all the mask ratios for both TAMP-Street and

DPED50k. Especially, our ITDiff outperforms LeftRefill with a large margin, *e.g.*, 3.6294%/1.3994%, 4.4814%/2.8310%, 5.3082%/4.4179%, 5.6756%/5.0150% PSNR improvements under four mask ratios respectively. ❷ Moreover, we notice the improvements on DPED50k are much greater than TAMP-Street which demonstrate the proposed TAMP task is harder than conventional reference-guided image inpainting task. In summary, the general improvements over existing methods proves the superiority of our ITDiff for solving TAMP. Please refer to Appendix E for visual comparison.

**Boosting Existing Methods with ITDE.** Since the proposed ITDE is a plug-and-play module, here we apply it to various existing methods as the second comparative experiment. As the results shown in Tab. 4, it is clear that our ITDE generally boosts existing methods for TAMP tasks. Notably, our ITDE consistently improves the performance for all mask ratios which proves the great image complementation capability of ITDE. To have a clear comparison, we also showcase the inpainting results of existing methods when integrated with ITDE. Fig. 6 shows the tested intact and damaged duo-image, with 20%-30% mask, noting we select it without significant content mismatch for exclusive ITDE boosting verification. As Fig. 7 illustrated, we can see that all three methods can only generate blurred or cluttered contents for large masked regions, *i.e.*, the red rectangles. With ITDE boosting, all of them can restore the original semantic structure as indicated in green rectangles. While the details are still insufficient, we show in Sec. 6.3 that our ITDiff can further refine them.

**Table 4:** The PSNR results after ITDE boosting on the TAMP-Street dataset. The best results are highlighted in **bold**.

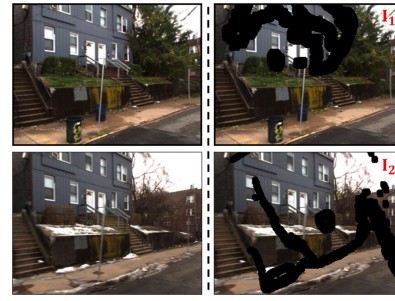

**Figure 6:** The exemplar intact and masked duo-image for ITDE boosting experiments.

| Method \ Mask | 20%-30% | 30%-40% | 40%-50% | 50%-60% |
|---|---|---|---|---|
| | TAMP-Street | | | |
| TransFill | 19.3377/20.1026 | 17.2356/17.6748 | 15.1677/15.8720 | 13.1019/13.7011 |
| **TransFill-ITDE** | **20.7309/21.4531** | **18.6612/19.0100** | **17.1427/17.6803** | **15.1152/15.2268** |
| TransRef | 18.9164/22.7751 | 17.2486/20.9556 | 15.9129/19.2969 | 14.6797/16.2012 |
| **TransRef-ITDE** | **20.5575/23.8705** | **18.9886/22.6911** | **17.4543/20.5932** | **16.8724/18.5080** |
| LeftRefill | 23.1727/23.1517 | 21.6623/21.9740 | 19.7019/19.5421 | 15.2239/15.9228 |
| **LeftRefill-ITDE** | **23.5043/24.9533** | **22.8759/23.9154** | **20.8503/20.4831** | **17.1051/17.9848** |
| | DPED50k | | | |
| TransFill | 21.5232/22.1119 | 18.5660/18.3195 | 17.7743/18.2417 | 15.6322/15.3806 |
| **TransFill-ITDE** | **22.8748/22.5024** | **20.1575/20.1024** | **18.9510/19.1646** | **17.7583/17.8872** |
| TransRef | 21.1612/26.6525 | 19.4944/24.6318 | 18.3366/22.8359 | 17.1906/19.4730 |
| **TransRef-ITDE** | **22.7150/26.9076** | **20.5920/25.4573** | **19.2140/23.8966** | **18.2249/20.8530** |
| LeftRefill | 26.5375/30.9959 | 23.8886/27.9177 | 21.5189/24.8386 | 17.8494/20.5472 |
| **LeftRefill-ITDE** | **27.1053/31.3443** | **25.4752/28.8902** | **22.1100/25.8750** | **19.9176/21.5294** |

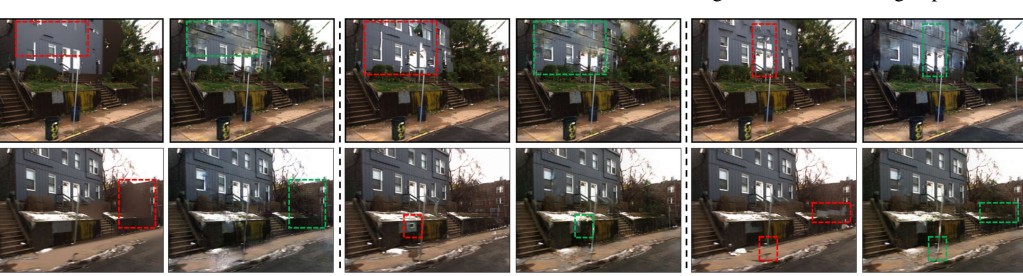

TransFill    TransFill-ITDE    TransRef    TransRef-ITDE    LeftRefill    LeftRefill-ITDE

**Figure 7:** The visualization of target duo-image inpainted by existing RefInpaint methods *v.s.* corresponding ITDE-boosted version. The defects and improvements are highlighted with red and green rectangles respectively.

**Reference-guided Image Inpainting *v.s.* TAMP.** We have shown ITDiff can achieve promising TAMP results, then it should also perform well for the conventional reference-guided image inpainting task since TAMP is a much harder task compared to it. As shown in Tab. 5, we evaluate our ITDiff and the baseline methods under the traditional reference-guided image inpainting setup. We can observe that our ITDiff also outperforms existing reference-guided image inpainting methods with a large margin under reference-guided image inpainting setup, which further proved the efficacy of our ITDE for realizing appropriate image complementation.

**Table 5:** Results of baselines and ITDiff for reference-guide image inpainting on DPED50k.

| Method \ Mask | 20%-30% | 30%-40% | 40%-50% | 50%-60% |
|---|---|---|---|---|
| TransFill | 25.3319 | 24.7700 | 22.2702 | 21.1901 |
| TransRef | 26.9667 | 24.8714 | 23.2505 | 20.0893 |
| LeftRefill | 30.9543 | 28.5670 | 26.4734 | 22.8566 |
| ITDE | **35.1201** | **31.0942** | **30.0931** | **27.9348** |

## 6.3 DISCUSSION

Throughout the preceding discussion, we did not explicitly elucidate how the proposed ITDE can address the challenges highlighted in the problem formulation, *i.e.*, Sec. 3.1. Indeed, we observe that

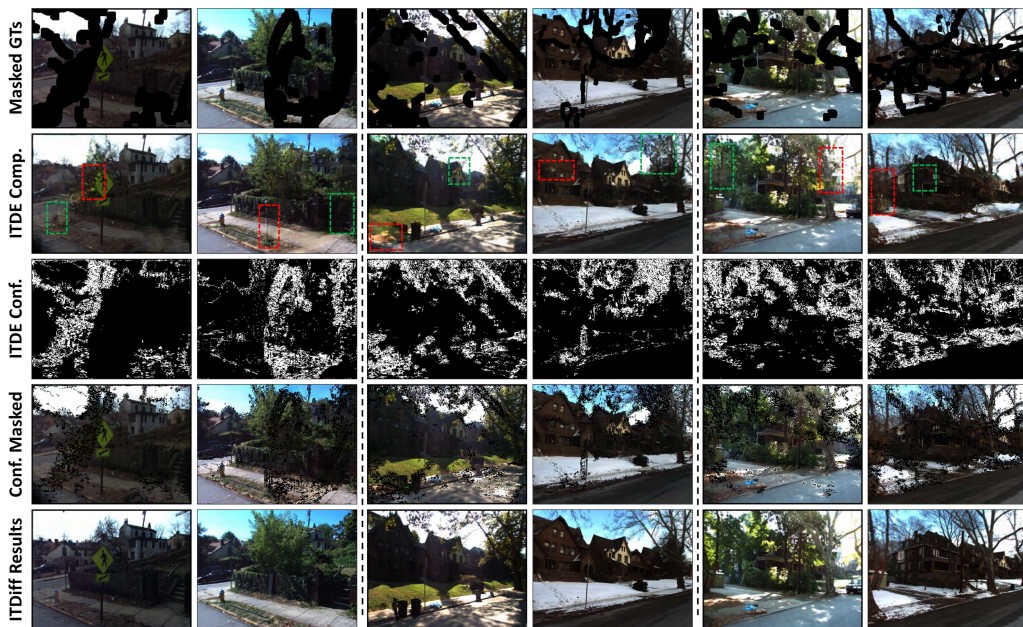

**Figure 8:** The visualization of ITDE and ITDiff outputs for three exemplar test duo-images. Note the ITDE confidence maps are inversed for better visualization. Better zoom in and view in color.

all the challenges essentially boil down to the same objective: mitigating the semantic discrepancy during image complement, *i.e.*, the insight derived at the beginning of Sec. 4. In this part, we demonstrate that our ITDE can achieve such desired image complementation by visualizing the interactive transition and contradictory content suppression capability of ITDE for the challenges.

**Interactive Transition Capability.** As "ITDE Comp." row shown in Fig. 8, our ITDE is capable of complementing images with information from the counterpart image, *i.e.*, the green rectangles. Especially, our ITDE does not introduce significant semantic contradictions during the complementation. While there are some undesirable contents still exist, *i.e.*, enclosed by red rectangles, we show in the next that the confidence map is capable of further suppressing such artifacts.

**Contradictory Content Suppression.** The confidence map indicates how reliable is the ITDE complemented results, based on which we realize the suppression of content contradiction. As shown in the "ITDE Conf." row in Fig. 8, it is clear that the confidence map highly correlated and refined the original mask. When we apply such confidence map as a mask to the ITDE complemented results, *i.e.*, "Conf. Masked" row, we see clearly that those undesired contents in "ITDE Comp." images are further suppressed. With the further utilization of the existing inpainting method, we can achieve much more reasonable results as shown in the "ITDiff Results" row in Fig. 8.

## 6.4 ABLATION STUDY

In this section, we ablate the proposed ITDiff to verify the contribution of different components. Note that the ablation is based on DDNM since the ITDiff main results, *i.e.*, Tab. 2, are achieved based on DDNM. In specific, we test four different settings, ❶ DDNM itself, ❷ DDNM with diffusion interaction, *i.e.*, DDNM-Interact, ❸ DDNM with ITDE boosting, *i.e.*, DDNM-ITDE, ❹ only ITDE. In summary, we observe that ITDE boosts the proposed ITDiff most under both 20%-40% and 40%-

**Table 6:** The ablation study results of proposed ITDiff on TAMP-Street dataset under 20%-40% and 40%-60% mask ratios.

| Method \ Mask | 20%-40% | 40%-60% | 20%-40% | 40%-60% |
|---|---|---|---|---|
| | PSNR ↑ | | $L_1$ ↓ | |
| DDNM | 22.8717/23.7006 | 18.9081/19.0504 | 0.0427/0.0386 | 0.0928/0.0921 |
| DDNM-Interact | 22.9135/23.7882 | 19.5523/19.6538 | 0.0401/0.0325 | 0.0891/0.0882 |
| DDNM-ITDE | 24.1619/24.9153 | 20.5487/21.1040 | 0.0253/0.0223 | 0.0414/0.0474 |
| ITDE | 23.0512/23.8046 | 19.4225/19.9832 | 0.0460/0.0429 | 0.0929/0.0889 |
| **ITDiff** | **24.1764/24.9189** | **20.5648/21.1180** | **0.0220/0.0211** | **0.0413/0.0424** |
| | SSIM ↑ | | LPIPS ↓ | |
| DDNM | 0.8755/0.8814 | 0.7526/0.7615 | 0.1493/0.1623 | 0.2832/0.3048 |
| DDNM-Interact | 0.8613/0.8732 | 0.7501/0.7593 | 0.1477/0.1583 | 0.2701/0.2900 |
| DDNM-ITDE | 0.8652/0.8883 | 0.7876/0.7699 | 0.1217/0.1368 | 0.2195/0.2294 |
| ITDE | 0.8627/0.8657 | 0.7323/0.7450 | 0.1222/0.1365 | 0.2304/0.2478 |
| **ITDiff** | **0.8953/0.8984** | **0.7878/0.7801** | **0.1117/0.1267** | **0.2192/0.2291** |

60% mask ratios. While the performance lifting of diffusion interaction is limited, it still contributes to the overall results. In general, our designs all contribute to the final performance, *i.e.*, ITDiff.

# 7 CONCLUSION

In this work, we present the Time-vAriant duo-iMage inPainting (TAMP) task which extends existing image inpainting research by covering the situation where two damaged images with a large time gap need to be restored simultaneously. We conduct intuitive experiments and find that even SOTA inpainting methods cannot achieve convincible results due to the failure of proper image complementation. Thus we design an explicit image complement module, *i.e.*, Interactive Transition Distribution Estimation (ITDE), for realizing effective image complementation process. The proposed ITDE is independent of inpainting pipeline thus we apply it to the SOTA diffusion model, *i.e.*, DDNM, and propose the Interactive Transition distribution-driven Diffusion (ITDiff) as our solution for TAMP. Furthermore, considering the lack of benchmarks for TAMP, we built the TAMP-Street dataset to evaluate the models with fairness. The experimental results on TAMP-Street and conventional DPED50k dataset demonstrate that our ITDE can generally boost existing methods for TAMP task.

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

## A  DATASET VISUALIZATION

We show some image and mask pair examples from TAMP-Street dataset in Fig. 9 and Fig. 10 respectively.

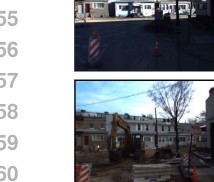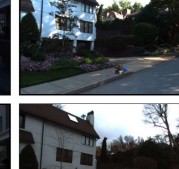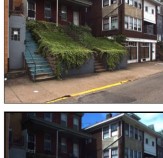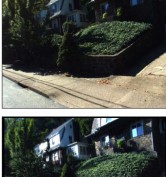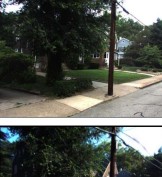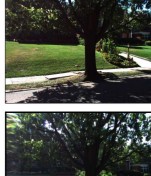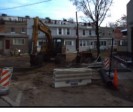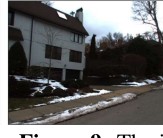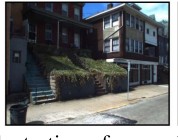

**Figure 9:** The illustration of example images for TAMP-Street dataset.

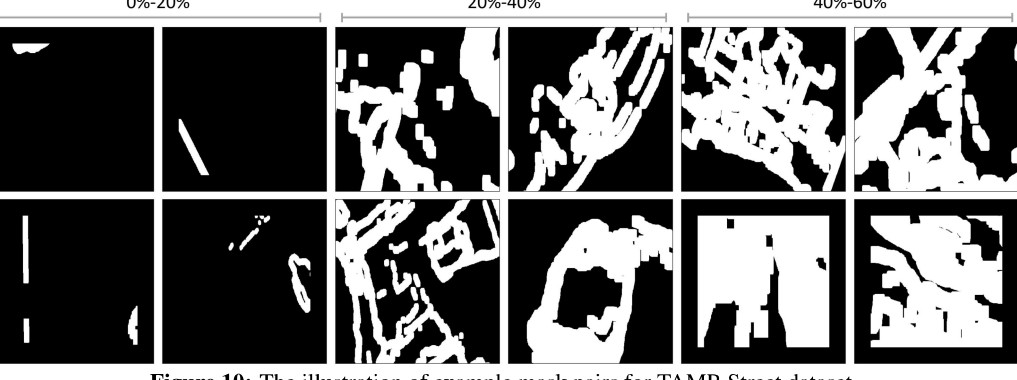

**Figure 10:** The illustration of example mask pairs for TAMP-Street dataset.

## B  COMPREHENSIVE EXPERIMENTAL SETTINGS

### B.1  EXPERIMENTAL ENVIRONMENT

We implement our code with Python 3.8.18 based on PyTorch 1.7.1 and cuda11. All the experiments are conducted on the same workstation with an AMD EPYC 7763 64-Core CPU, 504GB RAM, and four NVIDIA A40 GPUs (46GB memory each). The operating system is Ubuntu 20.04.

### B.2  INTUITIVE EXPERIMENT DETAILS

During the intuitive experiments, *i.e.*, Sec. 3.2, the official released code for LeftRefill [2] is utilized. We follow the instruction to train the model on TAMP-Street and DPED50k training splits respectively. Another point is that we keep resizing the input into 512x512 which is the size used during model pretraining of LeftRefill. The model output is further resized to the original size for final visualization. For all the other setups of intuitive experiments, we keep them the same as the officially released code. Two more duo-image results under the intuitive experiment setting are shown in Fig. 11

### B.3  MAIN EXPERIMENT DETAILS

**ITDE Training.**  We train ITDE on the training split of TAMP-Street and DPED50k datasets with the assembled training mask pairs. During training, both images and masks are resized into $256 \times 256$ and there is no data augmentation applied. The images are normalized into [-1,1] and the masks are binarized into 0 and 1. Following Li et al. (2022c), we set the weights for $L_1$, GAN, Style and perceptual losses as $L = L_1 + 0.1L_{GAN} + 250L_{Style} + 0.1L_{percetual}$. The discriminator of GAN

---

[2]https://github.com/ewrfcas/LeftRefill

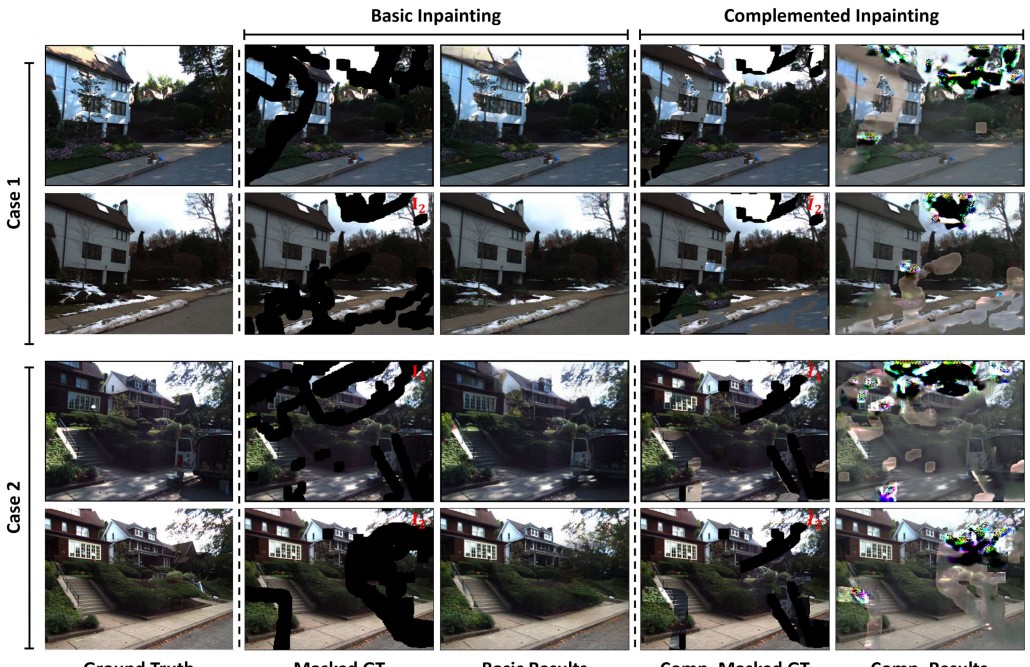

**Figure 11:** Another two duo-image inpainting results under intuitive experiment setup.

utilized during training is also set to the same as Li et al. (2022c). Specifically, we adopt Adam optimizer with $\beta_1 = 0, \beta_2 = 0.9$ and a learning rate $1 \times 10^{-4}$ to train ITDE for 200 epoch in total. We evaluate ITDE on the evaluation split of the two datasets and evaluation mask pairs every 5 epochs and save the model with the best PSNR as the pretrained ITDE for testing.

Note that both complement and confidence heads are trained together with the above setup. The only difference between the complement head and the confidence head is the ground truth for supervision. The ground truth for complement head training is the image from the dataset, while the ground truth for confidence head is the difference between ground truth and complement head output. That is to say, the confidence head is trained based on the complement head's outputs.

**Baseline Training.** To keep the fairness of comparison, all the baselines are pretrained on TAMP-Street and DPEd50k. Expect for TransFill (Zhou et al., 2021) which can only be evaluated by sending the data to their server for evaluation. As for the training details, we strictly follow each baseline's original released code.

## C  SMALL MASK RATIO RESULTS

As shown in Fig. 12, we test three different cases of small mask ratios for existing reference-guided image inpainting baselines and our ITDiff. It is clear that existing diffusion-based image inpainting methods are fully capable of realizing convincible results when the masked image region is small. Therefore, as we clarified in Sec. 5, we only adopt the masks with 20%-60% mask ratios for assembling the TAMP-Street dataset.

## D  LIMITATIONS

**Dataset.** At present, there is no dedicated benchmarks for evaluating TAMP task for which we assembled the TAMP-Street dataset. However, TAMP-Street dataset is relatively small in scale when compared with the benchmarks for other machine learning research tasks. Moreover, TAMP-Street mainly focuses on real-world urban street situations but there are many other practical real-world scenarios that can also fit into TAMP setup. Therefore, it would make TAMP more influential and practical by assembling large-scale datasets with sub-datasets for various real-world scenarios.

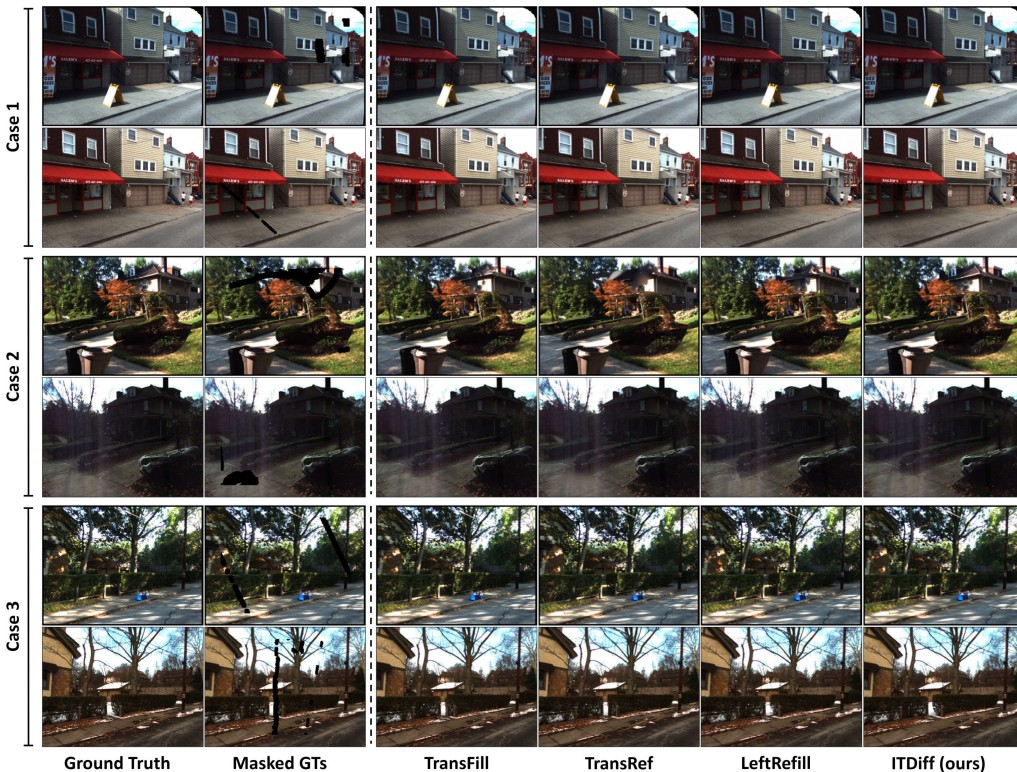

**Figure 12:** The illustration of SOTA reference-guided image inpainting methods and our ITDiff for inpainting target duo-images with small, *i.e.*, 0%-20%, mask ratio.

**Damage Setups.** In general, the TAMP task focuses on the situation where both target and reference images need to be restored at the same time. However, as one can expect, it is also of great practical application value by considering different image damage situations. For example, the two images suffer from different degree of pixel missing or both of them involves totally random degree of damage. Moreover, the pixel damages can also be extended to common corruptions Hendrycks & Dietterich (2019).

# E COMPARATIVE VISUALIZATION OF ITDIFF RESULTS

In this part, we show the comparative visualization results of conventional reference-guided image inpainting methods and our ITDiff for solving TAMP. As shown in Fig. 13, Fig. 14 and Fig. 15, we showcase three different TAMP situations for fair comparison. It can be observed that our ITDiff realizes the most reasonable results when compared with SOTA diffusion-based methods. Such visual quality boosting is much clearer when comparing our ITDiff results with traditional methods' outputs.

# F BROADER IMPACT

Generally, our work is limited to the research field of image restoration but still may be applied to various up-stream tasks, *e.g.*, cultural relic restoration and virtual scene editing. The TAMP task defined in this work is especially applicable for restoring images with time variance. Taking the image editing application as an example, our work reveals the real application of utilizing historical images for supervising the editing process and also can perform as an anti-malicious technique. In summary, this work not only introduced TAMP to cover more practical image inpainting scenarios, but also raised the concern of how should existing image restoration research reflect the real-world applications.

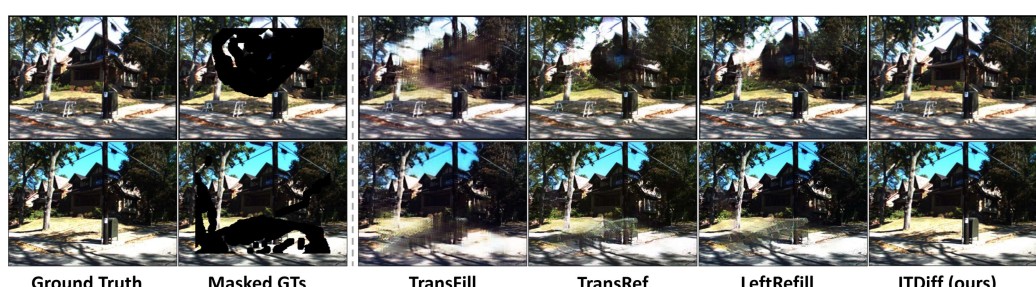

**Figure 13:** The first comparative visualization TAMP case inpainted by SOTA methods and our ITDiff.

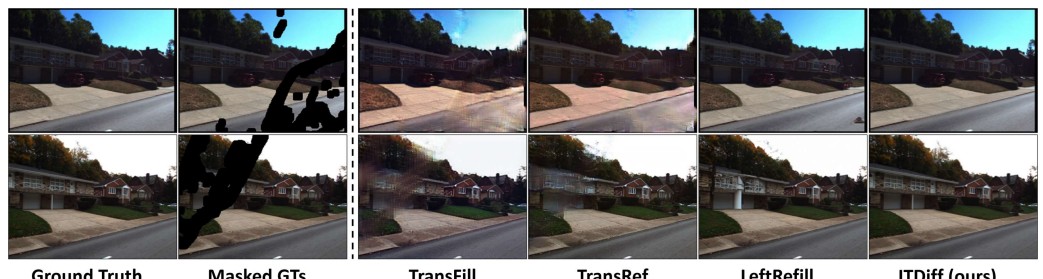

**Figure 14:** The second comparative visualization TAMP case inpainted by SOTA methods and our ITDiff.

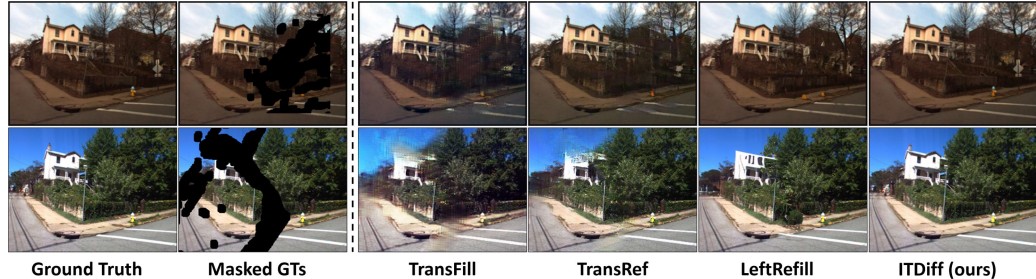

**Figure 15:** The third comparative visualization TAMP case inpainted by SOTA methods and our ITDiff.

