# OpenReview forum: "Time-variant Duo-image Inpainting via Interactive Distribution Transition Estimation"
_ICLR.cc/2025/Conference — Submitted to ICLR 2025_

### Official Review · Reviewer_zhtJ · 2024-10-23

**Soundness:** 3
**Presentation:** 3
**Contribution:** 3
**Rating:** 5
**Confidence:** 4

**Summary:**

This paper provides a subtask of reference-guided image inpainting, i.e., Time-vAriant duo-iMage inPainting (TAMP), with a TAMP-Street dataset and an Interactive Transition Distribution-driven Diffusion (ITDiff) model. The TAMP task mainly concerns the two challenges of object mismatch and appearance change for reference-guided image inpainting. The experiments were conducted on the TAMP-Street dataset compared with three reference-guided image inpainting methods.

**Strengths:**

1. The proposed reference-guided image inpainting task considers the reference images suffering from damage and having differences to the target images across the time.
2. The performance seems better compared to related reference-guided image inpainting methods.

**Weaknesses:**

The major contribution of the proposed method is the Interactive Transition Distribution Estimation (ITDE) module shown in Figure 5, while the performance seems better, here comes two issues for this module:
1. The novelty of the ITDE module compared to existing U-Net-style designs.
2. The efficacy of the ITDE module for boosting the performance.

**Questions:**

1. What are the key differences between the structure of ITDE module compared to previous similar designs (actually there are so many U-Net-based neural networks)? For example, there are many dual-branch U-Net-style neural network designs as follows: "Crossed Dual-Branch U-Net for Hyperspectral Image Super-Resolution" (IEEE Journal of Selected Topics in Applied Earth Observations and Remote Sensing, 2023), "DI-UNet: dual-branch interactive U-Net for skin cancer image segmentation" (Journal of Cancer Research and Clinical Oncology, 2023), "DBFU-Net: Double branch fusion U-Net with hard example weighting train strategy to segment retinal vessel" (PeerJ Computer Science, 2022), "Multi-Branch U-Net for Interactive Segmentation" (IEEE Signal Processing Letters, 2024), "Y-Net: Dual-branch Joint Network for Semantic Segmentation" (ACM Transactions on Multimedia Computing, Communications, and Applications, 2021), etc.
2. How to validate the performance boost mainly comes from the ITDE module (e.g., not because of the diffusion)?
3. How to illustrate the real efficacy of ITDE module exactly deal with the issue of semantic contradiction of complemented contents claimed in the paper?
4. What are the failure cases and potential applications of the work?

---

### Official Review · Reviewer_UTpG · 2024-10-27

**Soundness:** 3
**Presentation:** 3
**Contribution:** 2
**Rating:** 3
**Confidence:** 3

**Summary:**

This paper resolves an interesting task, TAMP, which inpaints paired damaged images from the same scene. Authors further collected a dataset to evaluate the effectiveness of solving TAMP. Experiments prove its SOTA effect.

**Strengths:**

1. This paper achieved information exchange between two images taken at different times in the same scene, which is novel.
2. This paper presented a new task, TAMP, and collected a dataset for evaluation.

**Weaknesses:**

1. I think TAMP itself is a fake task. First, it is specifically designed for two images of the same scene taken at different times, both of which are incomplete. In fact, if I want to restore an image of a scene, I can use Google Maps or other methods to capture the complete image of the scene to assist in completion, rather than using another incomplete image. The importance of this article is questionable. Second, I believe this method can benefit image inpainting task. Given a incomplete image, a diffusion model can be used to generate a complete scene like image, which allows for some distortion. Then use the method described in this article to complete the given incomplete image using the pre generated image. From this perspective, this method has some value, but I did not see any relevant discussion in the paper itself.
2. The operation of exchanging data within the diffusion model have been widely used. Although authors claimed that ITDE is their proposed novel module, it is actually a normal cross-attention operation that has been widely used in diffusion models. It is obvious that cross-attention can achieve data-exchange between two images. I believe the method itself lacks novelty.

Overall, TAMP is fake task but this method can benefit image inpainting. However, if we ignore this new task and only check the algorithm itself, it still lacks innovation.

**Questions:**

Why don't you apply this method to pure image inpainting to prove its practicability and effectiveness more clearly? Is the performance of this method not good enough compared to other image completion methods?

---

### Official Review · Reviewer_qjPR · 2024-10-31

**Soundness:** 4
**Presentation:** 3
**Contribution:** 3
**Rating:** 5
**Confidence:** 4

**Summary:**

In this paper, the authors propose a method to relax the constraints on reference images in the reference-guided image inpainting problem. They explain the rationale for developing their method based on the observation that the given reference image may be damaged or there may be a pixel-level domain gap between the reference image and the target image. To address these issues, they propose a neural network-based module called "Interactive Transition Distribution Estimation" (ITDE).  This module is trained to maintain semantic consistency between the duo images (reference image and target image) while providing refined input for subsequent image inpainting processes. Additionally, they integrated ITDE with state-of-the-art diffusion models to create an Interactive Transition Distribution-based Diffusion (ITDiff) model. To demonstrate its superiority, they conducted comparative experiments and ablation studies on both newly proposed datasets and existing representative datasets, comparing their method with existing approaches.

**Strengths:**

+ The authors explain the history of development in this field and clearly describe the limitations of existing methods. Through this process, they clearly define the problem they aim to solve. Furthermore, they not only mathematically model this problem precisely but also clearly explain their motivation and methodology.
  * They demonstrate from various view points that existing methods alone are insufficient to solve the problem.
+ The authors provided detailed explanations of their model and presented their experimental setup specifically. Additionally, they have released their dataset used in comparison experiments. This will be of great help to other researchers in this field in the future.
+ They demonstrate the practical significance of their claims through various experiments. The experiments strongly support their problem statement.

**Weaknesses:**

- The authors need to provide their actual results in higher quality. It is sometimes difficult to determine if the provided results are indeed better. I suggest including additional results with the original images in the supplementary material.
- The authors claim that their method can be used in a plug-and-play manner, but they do not explain why they didn't solve this problem end-to-end.
- I believe improvements are needed in the explanation of the modules proposed by the authors. For example, additional detailed explanations on how the two headers are combined and trained, and reorganization of the explanation about the loss function, etc.

**Questions:**

* Is the input image already corrupted, when the image and mask are input into the ITDE model? If so, what is the reason for providing a mask?
  * It would be beneficial to mention this situation additionally. This point is not clearly evident in Figure 5.
  * In real-world usage scenarios, does a user actually need to create the mask?
  * How does this module operate when the corrupted image and mask are not well-aligned?
* In Equation 3, how many surrounding pixels form the neighborhood? How does the performance of the module change with respect to this?
* Could you provide additional explanation on how the confidence head in ITDM is trained, and why we can trust this value?
* The authors claim that their method can be used in a plug-and-play manner, but they do not explain why they didn't solve this problem end-to-end. It would be good to include an explanation for this.

If the authors can address my issues in the rebuttal, I am willing to raise my score.

**Details Of Ethics Concerns:**

The authors address the limitations of their method and its broader impact in the supplementary material.

---

### Official Review · Reviewer_nHMH · 2024-11-04

**Soundness:** 2
**Presentation:** 2
**Contribution:** 2
**Rating:** 3
**Confidence:** 4

**Summary:**

The paper introduces a novel task termed 'time-variant duo-image inpainting,' which seeks to reconstruct an image using a reference image that exhibits substantial discrepancies and contains damage. The authors propose a method called ITDE to generate plausible content with associated confidence scores from the input images. These predictions are subsequently utilized by inpainting models to complete the image. Additionally, the authors have compiled a street scene dataset to serve as a benchmark for assessing the performance of their approach.

**Strengths:**

1. The authors have conducted a series of experiments, presenting both quantitative and qualitative analyses.
2. The quantitative results indicate that the proposed method outperforms existing approaches.
3. The paper compares the proposed method against a range of baselines, encompassing both GAN and diffusion-based models.

**Weaknesses:**

### Motivation

1. The rationale for introducing a time-variant duo-image inpainting task is unclear. It is not evident why this new task is necessary when reference-guided image inpainting already exists. The paper claims that significant misalignment is not considered in existing works (L41-42), but this assertion is disputed.
2. The practicality of reference images having missing regions for real-world applications is questionable.
3. The term "duo-image inpainting" is ambiguous and requires clarification.
4. The paper's core contribution appears to be a two-stage GAN-based model for predicting content and confidence from reference images, rather than a "plug-and-play" module as claimed. The authors should provide a clear distinction to prevent misunderstanding.

### Approach

1. The term "interactive" is misleading as it is not clear how the model interacts with users.
2. Key details are omitted, such as the optimization function for the proposed module and the supervision strategy for the confidence head.
3. The model's performance on the dataset shown in Figure 4 is unconvincing, as LeftRefill was not trained on this dataset. The authors should present results after full training on the same dataset.
4. The term "interactive transition distribution estimation" is confusing. The proposed module seems to focus on extracting convincing content from damaged reference images with confidence prediction, but the meaning of "interactive," "transition," and "distribution" is not apparent.

### Experiments

1. It is unclear whether all baselines were trained with the same data as the proposed method. The visual quality improvements of the proposed model, as shown in Figure 2, appear to be minimal.
2. Given that the proposed model is a two-stage framework for reference-image inpainting, an ablation study comparing the two-stage design to an end-to-end design is suggested to validate the effectiveness of the ITDE.

**Questions:**

My major questions are as follows,
1. Task Necessity and Definition: The justification for the introduction of a new task, specifically the "time-variant duo-image inpainting" task, is not compelling. The paper does not sufficiently differentiate this task from existing reference-guided image inpainting tasks, which also account for misalignments. The novelty and utility of this task need to be clearly articulated.

2. Model Complexity and Clarity: The paper misleadingly presents a two-stage model as a "plug-and-play" module. This oversimplification obscures the complexity of the model and may lead to misunderstandings about its integration and application in practical scenarios.

3. Reproducibility of Experimental Setup: The reproducibility of the experiments is questionable due to the random generation of inpainting masks. For a fair comparison, all baselines would need to be retrained under the same conditions, which is not addressed in the paper. This lack of standardization undermines the reliability of the comparative results.

Please find more details in Weaknesses.

---

> ### Comment · Reviewer_nHMH · 2024-11-26
> **after discussion, period, I decide to keep my original rating**
>
> Since the authors didn't provide any rebuttal during discussion period, I decide to keep my original rating.

---

### Meta-Review · Area_Chair_dJwg · 2024-12-20

**Metareview:**

In this paper, the authors presented a new task termed Time-vAriant duo-iMage inPainting (TAMP) that inpaint two damaged images from the same scene but captured at different times. This is a similar task as the reference-guided image inpainting but with different settings of the images used. An interactive transition distribution estimation module together with a state-of-the-art diffusion model was proposed to address the TAMP problem. A dataset called TAMP-Street was assembled to support the study. Experiments showed the effectiveness of the proposed method for the TAMP task. The strengths of this paper include:
- Extensive experiments to evaluate the methods both quantitatively and qualitatively.
- Good review of the literature and history of this field, and the proposed problem was clearly defined.
- The experimental analysis supports the claims made by the authors for their method.

The weaknesses of this paper are:
- The motivation and rationality of the proposed TAMP problem. The specified setting of the problem is questionable and may not be practical in real applications. Reviewers are also concerned about the necessity of such a task given the availability of reference-guided image inpainting, and the practicality of the required images for the task.
- The technical method was not clearly presented, with missing details and misleading claims (e.g. the plug-and-play).
- The novelty of the proposed method. Some claimed novel operations were questionable, e.g. the ITDS module.
- Issues in the experiments. The reviewers were concerning about the fairness of the comparison, missing ablation studies, significance of the results, etc.

Overall, there are quite a few major concerns about this paper. Given that no rebuttal was presented by the authors, these major concerns remain unsolved, leading to the reject decision.

**Additional Comments On Reviewer Discussion:**

There was no rebuttal provided by the authors, and no discussion between the authors and reviewers apart from some reviewers acknowledging the missing rebuttal. During the AC-reviewers discussion phase, discussions were initialed. Due to the lack of rebuttal from the authors, the concerns raised by the reviewers were not addressed, two reviewers lowered their ratings accordingly. In the end, this paper received 2 Reject and 2 borderline Reject. The final decision from the AC is based on the review comments and the consistent recommendation from the reviewers.

---

### Decision · Program_Chairs · 2025-01-22

Reject